# Associations between food insecurity in high-income countries and pregnancy outcomes: A systematic review and meta-analysis

**Zoë Bell**[1,2‡]*, **Giang Nguyen**[1,3‡]*, **Gemma Andreae**[1], **Stephanie Scott**[1,3], **Letitia Sermin-Reed**[1], **Amelia A. Lake**[3,4], **Nicola Heslehurst**[1,3]

**1** Population Health Sciences Institute, Faculty of Medical Sciences, Newcastle University, Newcastle Upon Tyne, United Kingdom, **2** Department of Nutritional Sciences, King's College London, London, United Kingdom, **3** Fuse, The Centre for Translational Research in Public Health, Newcastle Upon Tyne, United Kingdom, **4** School of Health and Life Sciences, Teesside University, Middlesbrough, United Kingdom

‡ These authors share first authorship on this work.
* zoe.bell@kcl.ac.uk (ZB); Giang.Nguyen@newcastle.ac.uk (GN)

**Data Availability Statement:** All relevant data are within the manuscript and its Supporting Information files.

**Funding:** This work was supported by the the Economic and Social Research Council (ES/Y007905/1 to ZB). The funders had no role in study

## Abstract

### Background

Maternal nutrition is crucial for health in pregnancy and across the generations. Experiencing food insecurity during pregnancy is a driver of inequalities in maternal diet with potential maternal and infant health consequences. This systematic review explored associations between food insecurity in pregnancy and maternal and infant health outcomes.

### Methods and findings

Searches included 8 databases (MEDLINE, Embase, Scopus, Web of Science, PsychInfo, ASSIA, SSPC in ProQuest, and CINAHL), grey literature, forwards and backwards citation chaining, and contacting authors. Studies in high-income countries (HICs) reporting data on food insecurity in pregnancy and maternal or infant health, from January 1, 2008 to November 21, 2023 were included. Screening, data extraction, and quality assessment were carried out independently in duplicate. Random effects meta-analysis was performed when data were suitable for pooling, otherwise narrative synthesis was conducted. The protocol was registered on PROSPERO (CRD42022311669), reported with PRISMA checklist (S1 File). Searches identified 24,223 results and 25 studies ($n = 93,871$ women) were included: 23 from North America and 2 from Europe. Meta-analysis showed that food insecurity was associated with high stress level (OR 4.07, 95% CI [1.22, 13.55], $I^2$ 96.40%), mood disorder (OR 2.53, 95% CI [1.46, 4.39], $I^2$ 55.62%), gestational diabetes (OR 1.64, 95% CI [1.37, 1.95], $I^2$ 0.00%), but not cesarean delivery (OR 1.42, 95% CI [0.78, 2.60], $I^2$ 56.35%), birth weight (MD −58.26 g, 95% CI [−128.02, 11.50], $I^2$ 38.41%), small-for-gestational-age (OR 1.20, 95%, CI [0.88, 1.63], $I^2$ 44.66%), large-for-gestational-age (OR 0.88, 95% CI [0.70, 1.12] $I^2$ 11.93%), preterm delivery (OR 1.18, 95% CI [0.98, 1.42], $I^2$ 0.00%), or neonatal intensive care (OR 2.01, 95% CI [0.85, 4.78], $I^2$ 70.48%). Narrative synthesis showed food insecurity was significantly associated with dental problems, depression,

design, data collection and analysis, decision to publish, or preparation of the manuscript.

**Competing interests:** The authors have declared that no competing interests exist.

**Abbreviations:** AOR, adjusted odds ratio; ARR, adjusted risk ratio; CI, confidence intervals; GDM, gestational diabetes mellitus; GWG, gestational weight gain; HIC, High-Income Countries; LGA, large-for-gestational-age; LMIC, low and middle-income country; MOOSE, Meta-analyses of Observational Studies in Epidemiology; NICU, neonatal intensive care unit; NOS, Newcastle–Ottawa scale; OR, odds ratio; PFOS, perfluorooctane sulfonate; PRISMA, Preferred Reporting Items for Systematic Reviews and Meta-Analyses; SGA, small-for-gestational-age; SNAP, Special Supplemental Nutrition Assistance Program; T2DM, type 2 diabetes mellitus; USDA HFSSM, United States Department of Agriculture Household Food Security Survey Module; WHO, World Health Organisation; WIC, Special Supplemental Nutrition Program for Women, Infants, and Children.

anxiety, and maternal serum concentration of perfluoro-octane sulfonate. There were no significant associations with other organohalogen chemicals, assisted delivery, postpartum haemorrhage, hospital admissions, length of stay, congenital anomalies, or neonatal morbidity. Mixed associations were reported for preeclampsia, hypertension, and community/resilience measures.

## Conclusions

Maternal food insecurity is associated with some adverse pregnancy outcomes, particularly mental health and gestational diabetes. Most included studies were conducted in North America, primarily the United States of America, highlighting a research gap across other contexts. Further research in other HICs is needed to understand these associations within varied contexts, such as those without embedded interventions in place, to help inform policy and care requirements.

## Author summary

### Why was this study done?

- In high-income countries (HICs), food insecurity is a major public health concern given its wide-reaching impact on diet, nutrition, physical and mental health.

- Pregnancy is a life course stage for additional nutritional demand and financial pressures, making women vulnerable to food insecurity.

### What did the researchers do and find?

- We used random effect meta-analysis and narrative synthesis to analyse the findings of studies reporting associations between food insecurity in pregnancy and maternal and infant health, in HICs.

- Pregnant women experiencing food insecurity were more likely to have high stress levels, mood disorder, anxiety, depression, gestational diabetes mellitus (GDM), and poor dental health compared with women who were food secure.

### What do these findings mean?

- Food insecurity may be a contributing factor to maternal health and well-being outcomes, particularly relating to GDM and mental health.

- The main limitation is that the evidence-base was primarily from the USA and Canada and further research is needed in other contexts to inform policy and care and address inequalities in maternal and child health.

## Introduction

Food insecurity is "the limited or uncertain availability of nutritionally adequate or safe foods and limited or uncertain ability to acquire foods in socially acceptable ways" [1]. Poverty, unemployment, and low income are key drivers of food insecurity in both low- and middle-income countries (LMICs) and high-income countries (HICs) [2]. Further, the double burden of malnutrition with increasing rates of overweight, obesity, and noncommunicable diseases is also shared. However, it is important to consider HICs independently due to the vast differences not only in the food system, food supply, and food environment, but for many HICs, food insecurity was relatively hidden until after the 2008 global financial crisis and parallel rise in charitable food aid [3]. The global recession that followed led to increased poverty rates and unemployment in HICs [4,5]. Social and health inequalities have been further exacerbated from the Coronavirus pandemic, war, and ongoing cost-of-living crises meaning increased costs of food and energy prices [6,7]. Concurrently, lack of routine population measurement and use of noncomparable measurements has prevented accurate understanding of prevalence of food insecurity [8]. A study reporting prevalence in HICs between 2014 and 2018 estimated the prevalence of moderate or severe food insecurity as 6.5%, although there was a wide distribution at country level ranging from around 3% to 15% [9]. The study also reported sociodemographic patterns in prevalence, with age, gender, household occupancy, and income being important. More recently, the United Nations estimated that food insecurity doubled from the start to the end of 2020, impacting 135 to 265 million people worldwide [10,11]. During that period, the Coronavirus pandemic highlighted clear inequalities in access to food and health outcomes both between and within countries [12]. Ongoing since 2021, HICs have been experiencing a cost-of-living crisis, further pushing households experiencing financial hardship into poverty [7].

In HICs, food insecurity is a major public health concern given its wide-reaching significant impact on diet, nutrition, physical and mental health [13]. It is therefore important to understand the impact of experiencing food insecurity during critical periods of nutrition and development crucial for health across the generations [14]. One critical period is during the first 1,001 days of life (from conception until a child is 2 years of age), particularly during pregnancy. Suboptimal nutrition during pregnancy can contribute to poorer maternal and infant health outcomes [14]. During pregnancy and breastfeeding, women have more complex nutritional requirements, with increased potential health impacts from food insecurity [15,16]. Further, women are particularly vulnerable to experiencing food insecurity due to increased likelihood of working lower-paid, part-time jobs while maintaining caring responsibility of children and elderly within the family unit and/or heading a single parent household [17–19]. Within households, experiences of food insecurity may be unevenly distributed with women more affected than men [20]. Qualitative research shows women tend to be household food managers, thus experiencing the psychosocial costs from negotiating access to sufficient healthy food, often foregoing food to ensure others in their household are fed [21–23]. This increase in women's susceptibility to food insecurity could have consequences for pregnancy physical and mental health outcomes. Qualitative research reports maternal changes to food behaviours such as restricting eating patterns, reducing portion size, skipping meals, and foregoing food. Further, a recent review exploring associations between maternal food insecurity and pregnancy weight and diet reported poorer diet quality such as reduced consumption of fruit and vegetables, increased red processed meats, and lower intakes of vitamin E, and significantly increased odds of maternal obesity [24]. Maternal obesity is highly prevalent in HICs and increases pregnancy complications such as large-for-gestational age (LGA) babies, preeclampsia, gestational diabetes mellitus (GDM), and in the longer-term type 2 diabetes

[25,26], increased depression [27,28]. The recent review did not report associations between food insecurity and other pregnancy outcomes. Observational studies have found maternal food insecurity to be associated with poor maternal mental health and be a risk factor for adverse pregnancy and birth outcomes [29–31]; however, there is an absence of systematic reviews reporting meta-analysis of these associations in HICs contexts.

This systematic review aimed to explore the relationship between food insecurity and pregnancy outcomes relating to maternal physical and mental health, and infant health. The review is set within the context of the global recession from 2008, when HICs experienced increasing prevalence of poverty and food insecurity.

## Methods

The systematic review was registered on PROSPERO (CRD42022311669) and is reported as per the Preferred Reporting Items for Systematic Reviews and Meta-Analyses (PRISMA) guidelines [32] (S1 File) supplemented with MOOSE (Meta-analyses Of Observational Studies in Epidemiology) Checklist guidelines specific to observational studies [33] (see S1 File).

Database searches were completed November 21, 2023. A comprehensive search strategy was developed using search terms relating to "food insecurity" (exposure), "pregnancy" (population), and "observational" (study design) with Boolean operators (see S1 Table). We did not include search terms relating to outcomes, as the aim was to be holistic and include studies reporting any pregnancy outcomes specific to maternal and infant health. No search terms were included relating to HICs, and studies in LMIC contexts were excluded during the screening stages. The search strategy was applied to MEDLINE, Embase, PsycInfo, Scopus, Web of Science, Applied Social Science Index, and Abstracts (ASSIA), Social Sciences Premium Collection (SSPC) in ProQuest and Cumulative Index to Nursing and Allied Health Literature (CINAHL) from January 1, 2008 to November 21, 2023. To limit the potential impact of publication bias, additional searches were conducted with searches for grey literature carried out using Trove, Open Access Theses and Dissertations (OATD) and stakeholder websites (see S2 Table). We also conducted forward and backwards citation chaining for all included studies using citationchaser (an R package and Shiny app [34]). We contacted authors when additional data was required to inform the screening or to include in meta-analysis (see S3 Table). Supplementary searches and contacting authors were completed in March 2024.

A modified version of the Population, Intervention, Control, Outcome, Study design (PICOS) framework: Population, Exposure, Comparison, Outcome, Study design (PECOS) [35] was used to develop eligibility criteria. The population (P) was pregnant women living in HICs (GNI per capita ≥$12,696 [36]). The exposure (E) was an explicit measure of food insecurity, and the comparison group (C) was food security/no food insecurity. Eligible study designs (S) were quantitative observational studies and primary observational data sources from grey literature. A date restriction was applied and aligned with the global financial crises which occurred in 2008. We excluded any studies where the data were primarily collected prior to 2008. There were no restrictions on reporting language to limit bias [37].

The search results were deduplicated in EndNote 20 [38] and Rayyan [39]. Title and abstract screening were conducted in Rayyan by a team of 7 reviewers independently and in duplicate. Full-text screening for any potentially eligible studies was also carried out independently and in duplicate. Conflicts between reviewers screening decisions were resolved by group discussion. We contacted corresponding authors when required, including following up on published conference abstracts for any subsequent publication of full papers, or when there was missing information in the publication essential for the screening decisions (e.g., data collection time period). Data extraction and quality assessment were carried out independently in

duplicate, with conflicts resolved by group discussion. The data extraction template included: type of publication, study type, country of study, data collection time period, study aim, measurement and definition of food insecurity, study participant information, and study outcomes. The Newcastle–Ottawa scale (NOS) cohort study tool was used to assess the quality of the included studies [40]. This scale assesses information and selection bias, for example, relating to participant identification, representativeness, and measurement of exposures and outcomes, and confounding variables which are the key factors to consider when assessing quality and bias in observational study designs. A quality scoring system was applied based on the number of "stars" awarded: 0 to 2 was categorised as low quality, 3 to 5 stars medium quality, and 6 to 8 stars high quality (see S4 Table).

Meta-analysis was carried out when there were at least 3 studies reporting data suitable for pooling. If odds ratios (ORs) were not reported in the included papers, then they were calculated using reported frequency data or by contacting authors for the required data, with the aim of increasing the number of studies to pool in a meta-analysis. Summary ORs were calculated using the random effects model using Stata/MP version 18 [41,42]. Heterogeneity between studies was explored using $I^2$ [43], and >75% was considered significant heterogeneity [44]. The protocol included a plan to conduct meta-regression, sensitivity analysis, Eggers tests, and funnel plots for all meta-analyses; however, as no meta-analysis included 10 or more studies, these were not conducted [45]. Narrative synthesis was performed when meta-analysis was not possible due to too few studies reporting outcome data similar enough to pool, or when it was not possible to convert data reported to ORs [46]. Data were tabulated and grouped into thematic categories relating to maternal and infant health outcomes (a priori categories). Further data-driven thematic categories were developed based on the specific outcomes reported in the included studies (e.g., birth weight) and how these data were presented (e.g., birth weight for gestational age z-score). Narrative summaries accompany the tables to report associations between food insecurity in pregnancy and health outcomes.

## Results

### Search results

There were 24,223 results from the database searches: 11,377 were duplicates, 12,633 were excluded at the title and abstract screening stage, and 213 full texts were screened (Fig 1). There were 2,422 additional records identified from supplementary searches (543 from websites and 1,879 from forwards and backwards citation chaining). Details of contact with study authors are provided in S3 Table.

### Characteristics of included studies

There were 25 studies that met the inclusion criteria (Table 1 and see S5 Table). Twenty-two included studies [29–31,48–66] were peer-reviewed journal articles and 3 [67–69] were masters or PhD theses. All were cohort studies published between 2012 and 2023: 15 were prospective cohort studies [29,48,49,51–57,60,62,63,66,69] and 10 retrospective cohort studies [30,31,50,58,59,61,64,65,67,68]. Nearly all studies were conducted in North America, with 20 studies conducted in the United States of America [29,31,49–60,62–67], 3 studies conducted in Canada [30,68,69], and 1 study in France [48] and the United Kingdom [61]. Included study sample sizes ranged from 70 [51] to 21,080 [65] women (pooled sample size $n$ = 93,871; median 858, IQR 510 to 3,592). Food insecurity was primarily determined using the USDA Food Security Survey Module ($n$ = 18 studies) [29,30,48,50–53,55,58,60–66,68,69], although varied criteria from 1 to 18 items applied [70]. Other studies used single or two-item questions from a Pregnancy Nutrition Questionnaire [56], a rapid assessment tool for prenatal care [67],

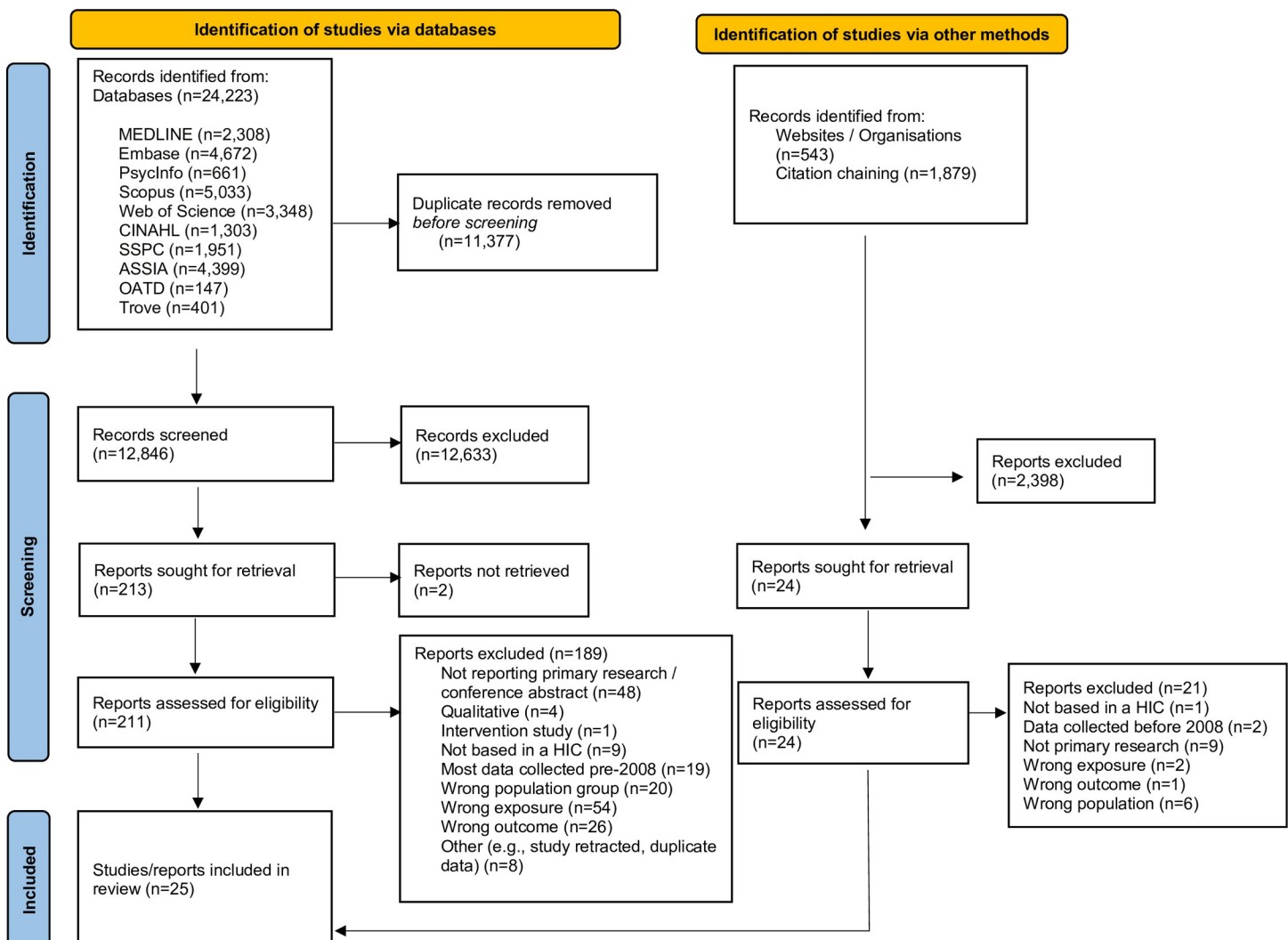

**Fig 1. Adapted PRISMA flow diagram showing the study selection process** [47]. ASSIA, Applied Social Sciences Index and Abstracts; SSPC, Social Sciences Premium Collection; CINAHL, Cumulative Index to Nursing and Allied Health Literature; OATD, Open Access Theses and Dissertations; HIC, High-Income Countries.

the Hunger Vital Sign ($n = 2$) [49,57], Pregnancy Medical Home risk screening [31], an undefined questionnaire in the Chemicals in Our Body study [54] and a single food insecurity question that specifically ask for the pregnancy period: "Does not having enough money make it hard for you to eat healthy food during pregnancy?" [59]. All included studies achieved between 4 and 8 stars in the quality assessment; 17 were rated as high quality and 8 medium quality (Table 1). All 25 studies were awarded a star for their representativeness of the exposed (food insecure) and non-exposed (food secure) cohorts (Q1 and 2), assessment of the exposure (Q3) and the duration of follow up (Q6). Nineteen (76%) were awarded a star for the comparability of cohorts on the basis of the design or analysis (Q4), and 18 (72%) for assessment of outcome (Q5). The lowest scoring element related to the adequacy of follow up (Q7; 44%, $n = 11$) indicating that loss to follow up and/or missing outcome data was a key limiting factor for study quality (see S6 Table).

While overall the studies reflected general maternity populations with their inclusion criteria, albeit usually singleton pregnancies and >18 years of age, there were some studies that

**Table 1. Summary of the included studies.**

| Included study (author, year) | Country of study | Study period | Study design | Total sample size | Food insecurity tools used | Quality assessment rating | Maternal physical health | Maternal mental health | Infant health |
|---|---|---|---|---|---|---|---|---|---|
| Richards and colleagues, 2021 [62] | USA | 2009–2014 | Prospective | 592 | USDA HFSSM 6 items | High | x | | |
| Luke, 2017 [67]* | USA | 2012–2013 | Retrospective | 9,472 | 1 question rapid assessment tool | High | x | x | x |
| Grilo and colleagues, 2015 [56] | USA | 2008–2011 | Prospective | 881 | 1 screening question | Medium | | x | x |
| Mak, 2019 [68]* | Canada | 2005–2014[1] | Retrospective | 4,817 | USDA HFSSM 10 items | High | | x | |
| Cheng and colleagues, 2022 [49] | USA | 2016–2017 | Prospective | 858 | Hunger vital sign 2 screening items | High | x | x | |
| Mehta and colleagues, 2020 [60] | USA | 2011–2013 | Prospective | 98 | USDA HFSSM 10 items | Medium | x | | |
| Eick and colleagues, 2020 [54] | USA | 2014–2018 | Prospective | 510 | 1 screening question | High | | x | |
| Lairara and colleagues, 2022 [58] | USA | 2010–2012 | Retrospective | 14,274 | USDA HFSSM 6 items | High | | x | |
| Goin and colleagues, 2021 [55] | USA | 2014–2018 | Prospective | 510 | USDA HFSSM | High | x | | x |
| Tarasuk and colleagues, 2020 [30] | Canada | 2005–2012[1] | Retrospective | 1,998 | USDA HFSSM 18 items | High | x | | x |
| Sandoval and colleagues, 2021 [64] | USA | 2018–2019 | Retrospective | 268 | USDA HFSSM 6 items | High | x | | x |
| Sullivan and colleagues, 2021 [29] | USA | 2011–2019 | Prospective | 541 | 3 selected items from the USDA HFSSM | Medium | | x | x |
| Power and colleagues, 2017 [61] | UK | 2007–2010 | Retrospective | 1,280 | USDA HFSSM 18 items | High | | x | |
| Tucker and colleagues, 2015 [31] | USA | 2011–2012 | Retrospective | 15,428 | 1 screening question | Medium | x | | x |
| Richards and colleagues, 2020 [63] | USA | 2009–2014 | Prospective | 746 | USDA HFSSM 6 items | High | | x | |
| Cooper and colleagues, 2022 [51] | USA | 2018–2019 | Prospective | 70 | USDA HFSSM | High | x | | x |
| Testa and colleagues, 2022 [65] | USA | 2016–2019 | Retrospective | 21,080 | 1 item from USDA HFSSM | High | x | | |
| Cheu and colleagues, 2020 [50] | USA | 2018 | Retrospective | 299 | USDA HFSSM 10 items | High | x | | x |
| Bihan and colleagues, 2023 [48] | France | 2012–2018 | Prospective | 1,168 | 1 single item adapted from USDA HFSSM** | Medium | | | x |

(*Continued*)

**Table 1.** (Continued)

| Included study (author, year) | Country of study | Study period | Study design | Total sample size | Food insecurity tools used | Quality assessment rating | Maternal physical health | Maternal mental health | Infant health |
|---|---|---|---|---|---|---|---|---|---|
| Shriver and colleagues, 2023 [66] | USA | 2019–2020 | Prospective | 299 | USDA HFSSM 6 items | Medium | x | | |
| Eagleton and colleagues, 2023 [53] | USA | 2019–2020 | Prospective | 170 | USDA HFSSM 6 items | High | | x | x |
| Oresnik, 2020* [69] | Canada | 2017–2018 | Prospective | 3,592 | USDA HFSSM 18 items and 1 single item adapted from USDA HFSSM*** | High | x | x | |
| Joseph and colleagues, 2023 [57] | USA | 2019–2021 | Prospective | 1,608 | Hunger vital sign 2 screening items | High | | | x |
| Meeker and colleagues, 2023 [59] | USA | 2020 | Retrospective | 12,525 | 1 single item from USDA HFSSM**** | Medium | | x | |
| Duh-Leong and colleagues, 2023 [52] | USA | 2019–2021 | Prospective | 787 | USDA HFSSM 5 items | Medium | | | x |

[1]Despite containing data collected before 2008, this study was included as the majority of the study period was post-2008.

*Master dissertation/ PhD thesis.

**Food insecurity was assessed using a single item "Which of these statements best describes the food eaten in your household? (i) enough of the kinds of food you want to eat; (ii) enough but not always the kinds of food you want to eat; (iii) sometimes not enough to eat; (iv) often not enough to eat."

***Food insecurity was assessed using a single item: "Does not having enough money make it hard for you to eat healthy food during pregnancy?"

****Food insecurity was assessed using a single item: worried whether the food would run out.

UK, United Kingdom; USA, United States of America; USDA HFSSM, USDA Household Food Security Scale Model.

applied criteria that may have excluded food insecure populations (S5a Table). These included restrictions to English speaking populations [29,50,53,60,66], exclusion of women in prisons [62,63,68], or First Nations reserves and settlements, Armed Forces, or living in care homes [68]. There were various and inconsistent participant characteristics reported by the included studies including mean, median, or categories of maternal age ($n = 15$); ethnic group or race ($n = 25$); parity, previous pregnancy, or having other children ($n = 15$); income brackets, quintiles, poverty line, and income need ratio ($n = 15$); various categories of educational attainment ($n = 21$); employment status ($n = 9$); relationship status ($n = 16$); and other socioeconomic indicators including measures of social security, WIC use, financial support, and housing tenure, occupancy, and neighbourhoods ($n = 16$) (see S5b Table). The mean or median age ranged from 18.6 to 33 years. Ethnic and racial representation included white, black, Hispanic, Latina, Asian, Pacific Islander, Pakistani, Bangladeshi, Indian, American Indian, Alsakan, and Indigenous groups. Majority populations included in the studies were white ($n = 10$ studies), Hispanic or Latina ($n = 5$), black ($n = 2$), South Asian ($n = 1$), or equal distribution between 2 or more groups ($n = 5$). Education less than high school equivalent ranged from 8.7% to 34.7% of participants. Women were more likely to have had previous pregnancies and/or other children in 9 studies, and a first pregnancy in 4 studies. There were more studies reporting that their participants were either married or in a relationship ($n = 13$) than unmarried or single ($n = 3$). More studies also reported a higher proportion of their participants to be in employment ($n = 6$) than not employed ($n = 3$).

## Pregnancy outcomes

All included studies reported associations between food insecurity and maternal health outcomes (*n* = 13 physical health [30,31,49–51,55,60,62,64–67,69] and *n* = 12 mental health [29,49,53,54,56,58,59,61,63,67–69]) and *n* = 13 infant health outcomes [29–31,48,50–53,55–57,64,67]. There were 17 included studies that reported adjusted models for some or all their analysis [30,49,51–55,57–59,62–65,67–69]. There was variation in variables included in adjusted models. Those most frequently included were maternal age and race or ethnicity (*n* = 15 each), followed by education (*n* = 13) and income (*n* = 10). Less frequently included were factors relating to housing or relationship status (*n* = 8 each), health insurance (*n* = 6), maternal BMI, previous pregnancy, clinical or research design factors (*n* = 5 each), WIC use or language spoken (*n* = 2 each), and immigration status or social support (*n* = 1 each).

## Food insecurity and maternal physical health in pregnancy

Outcomes reported were related to GDM (*n* = 9) [29,49–51,57,62,66,67,69], hypertensive disorders (*n* = 5) [29,30,48,57,67], mode of delivery (*n* = 4) [48,50,51,57], oral health (*n* = 1) [65], serum concentrations of organohalogen chemicals (*n* = 1) [60], postpartum haemorrhage (*n* = 1) [50], hospital admissions (*n* = 1) [30], and maternal morbidity (*n* = 1) [57] (S7 Table). Meta-analysis was possible for GDM and cesarean delivery. Nine studies reported associations between food insecurity and GDM and 8 were pooled in the meta-analysis (see S7 Table) [29,49–51,62,66,67,69]. This included 6 high-quality studies [49–51,62,67,69] and 2 medium quality [29,66], with a pooled sample size of 15,408 women. There was a significantly increased odds of GDM among pregnant women who experienced food insecurity compared with those that did not (OR 1.64, 95% CI [1.37, 1.95], $I^2$ 0.0%) (Fig 2). The one study not included in the meta-analysis reported no association between food insecurity and GDM (ARR 1.16, 95% CI [0.76, 1.77]) [57]. One study additionally reported no association between food insecurity and isolated hyperglycaemia (AOR 0.89, 95% CI [0.42, 1.89]) or impaired glucose tolerance (AOR 1.04, 95% CI [0.36, 3.02]) [49].

Three studies reported associations between food insecurity and cesarean delivery, and all were pooled in a meta-analysis (Fig 3). This included 2 high-quality studies [50,51] and 1 medium quality [48] with a pooled sample size of 1,256 women. Pooled analysis showed no association between food insecurity and cesarean section (OR 1.42, 95% CI [0.78, 2.60], $I^2$ 56.35%) (Fig 3).

Narrative synthesis (see S7 Table) identified no consistent pattern in direction of effect for food insecurity and preeclampsia: 1 study reported significant association (OR 1.91, 95% CI [1.11, 3.29]) and 2 others did not (OR 1.78, 95% CI [0.77, 4.10] and ARR 0.91, 95% CI [0.8, 1.03] [for preeclampsia without severe features or gestational hypertension] and ARR 0.95, 95% CI [0.78, 1.16] [for preeclampsia with severe features]) [29,48,57]. Results for chronic or gestational hypertension also showed an inconsistent association with food insecurity, with 1 study reporting a positive association (OR 16.94, 95% CI [7.79, 36.85]) [48] and 2 others showed no association (OR 1.45, 95% CI [0.65, 3.24] [29] and OR 1.22, 95% CI [0.94, 1.59]) [67]. One study reported a combined outcome of hypertension or diabetes which showed no association with marginal or moderate/severe food insecurity compared with food security (ARR 0.99, 95% CI [0.51, 1.92] and 1.44, 95% CI [0.88, 2.35], respectively) [30].

One study [65] reported significantly increased odds for a variety of oral health and dental care needs among women experiencing food insecurity, such as needing to see a dentist for a problem, not knowing it was important to care for teeth, and not talking about dental health with providers (AORs ranged from 1.29, 95% CI [1.04, 1.59 to 1.91], 95% CI [1.62, 2.25]). One study reported a positive association between serum perfluorooctane sulfonate (PFOS)

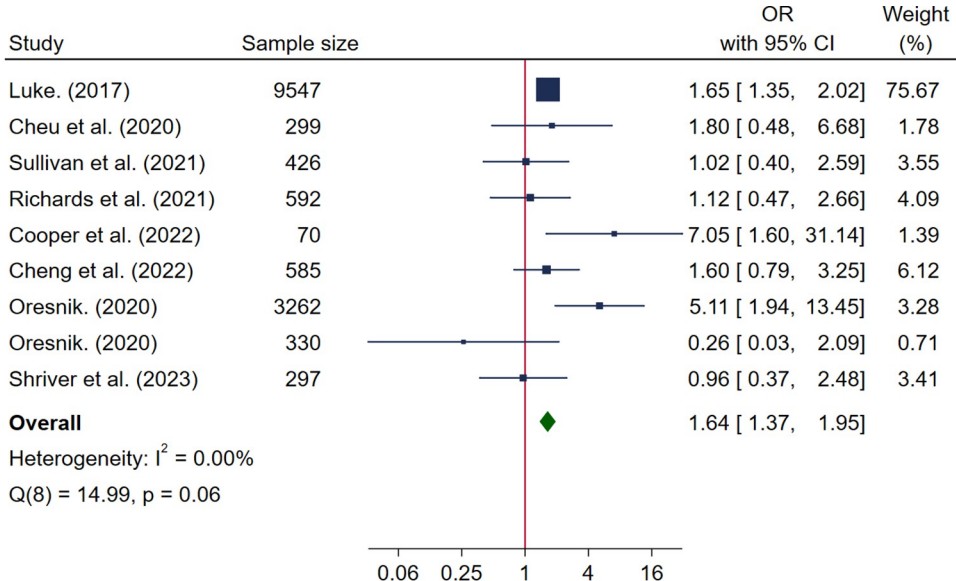

**Fig 2. Forest plot for meta-analysis of the association between gestational diabetes mellitus and food insecurity.** Legend: Categories of food insecurity and food security: food secure vs. food insecure, Luke, 2017 [67], Sullivan and colleagues, 2021 [29], Cheng and colleagues, 2022 [49], Cooper and colleagues, 2022 [51], Orsenik, 2020 [69], Shriver and colleagues, 2023 [66]; fully food secure vs. marginally food secure/food insecure, Richards and colleagues, 2021 [62]; adequate food security vs. inadequate food security, Cheu and colleagues, 2020 [50]. Measurement of FI: Luke, 2017 used 1 question rapid assessment [67]; Cheu and colleagues, 2020 used USDA HFSSM 10 items [50]; Sullivan and colleagues, 2021 used 3 items from USDA HFSSM [29]; Richard and colleagues, 2021 used USDA HFSSM 6 items [62]; Cheng and colleagues, 2022 used the Hunger vital sign 2 screening items [49]; Cooper and colleagues, 2022 used USDA HFSSM 18 items [51]; Oresnik, 2020 used USDA HFSSM 18 items and 1 single item adapted from USDA HFSSM [69]; Shriver and colleagues, 2023 used USDA HFSSM 6 items [66].

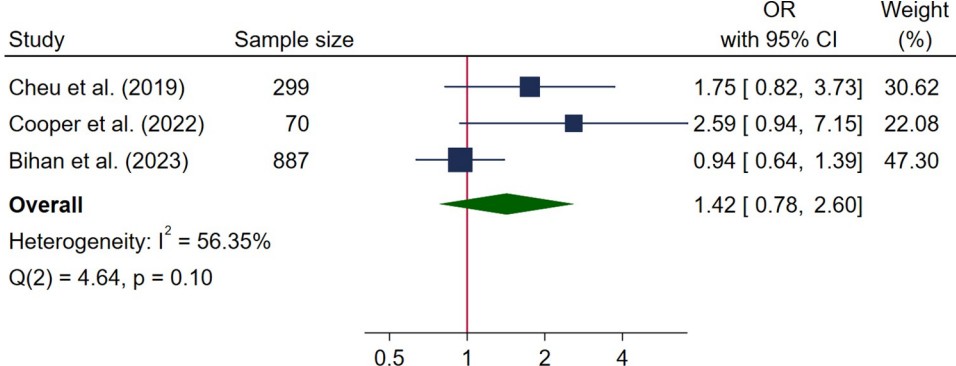

**Fig 3. Forest plot for meta-analysis of the association between cesarean section and food insecurity.** Legend: Categories of food insecurity and food security: adequate food security vs. inadequate food security, Cheu and colleagues, 2020 [50]; food secure vs. food insecure, Cooper and colleagues, 2022 [51]; food insecurity vs. not food insecurity, Bihan and colleagues, 2023 [48]. Measurement of FI: Cheu and colleagues, 2020 used USDA HFSSM 10 items [50]; Cooper and colleagues, 2022 used USDA HFSSM 18 items [51]; Bihan and colleagues, 2023 used 1 single item adapted from USDA HFSSM [48].

concentrations among women with low/very low food security compared with marginal/high food security (β 0.27, 95% CI [0.04, 0.50]), but no association with any other organohalogen chemicals investigated [60]. Additional outcomes reported with no association with food insecurity were [50,51] postpartum haemorrhage (OR 1.80, 95% CI [0.48, 6.62]) [50], severe maternal morbidity (ARR 1.05, 95% CI [0.69, 1.60]) or non-transfusion severe maternal morbidity (ARR 1.12, 95% CI [0.60, 2.12]) [57], or assisted delivery (OR 0.96, 95% CI [0.11, 7.98]) [50]. There was no significant association or consistent pattern in direction of effect for food insecurity and hospital admissions or length of stay in one study [30].

## Food insecurity and maternal mental health outcomes

Eleven studies reported associations between food insecurity and maternal mental health outcomes including depression (n = 3) [49,54,63], depressive symptoms (n = 4) [53,56,58,59], stress (n = 4) [49,54,67,68], anxiety-related outcomes (n = 6) [49,53,56,59,68,69], and other disorders (n = 4) [50,61,68,69], as well as outcomes relating to mental health (n = 4) [29,53,54,68] (see S8 Table). Three studies reported data that could be pooled for high levels of stress [49,67,68] and mood disorder [50,68,69]. All studies were high quality with a pooled sample size of 19,440 women for high stress and 12,588 women for mood disorders. Pooled analysis showed food insecurity was positively associated with high levels of stress with high heterogeneity (OR 4.07 95% CI [1.22, 13.55], $I^2$ 96.40%) (Fig 4) and mood disorder (OR 2.53 95% CI [1.46, 4.39], $I^2$ 55.62%) (Fig 5). One study reported additional data on food insecurity and experiencing 1–2 and 3–5 stress events, which were both significantly increased (OR 3.11, 95% CI [2.21, 4.37, and 12.46], 95% CI [8.99, 17.27], respectively) [67] (see S8 Table). One study also reported no association between perceived stress and food insecurity (β 0.52, 95% CI [−0.17, 1.33]) [54].

Narrative synthesis was conducted for the remaining mental health outcomes (see S8 Table). Three studies reported data for depression [49,54,63]. Two showed a positive association with diagnosis of depression among women experiencing food insecurity compared with food secure (AORs 3.68, 95% CI [1.43, 9.43, and 4.42], 95% CI [2.33, 8.35]) [49,63]. One study showed a positive association between food insecurity and depression scale score (β 2.67, 95%

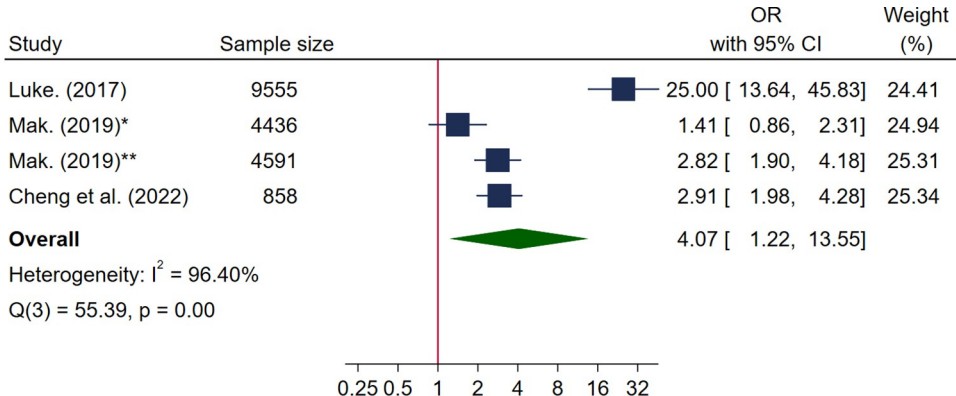

**Fig 4. Forest plot for meta-analysis of the association between high stress and food insecurity.** Legend: Categories of food insecurity and food security: food secure vs. food insecure, Luke, 2017 [67], Cheng and colleagues, 2022 [49]; food secure vs. marginal food insecurity* and for moderate or severe food insecurity**, Mak, 2019 [68]. Measurement of FI: Luke, 2017 used 1 question rapid assessment [67]; Cheng and colleagues, 2022 used the Hunger vital sign 2 screening items [49]; Mak, 2019 used USDA HFSSM 10 items [68].

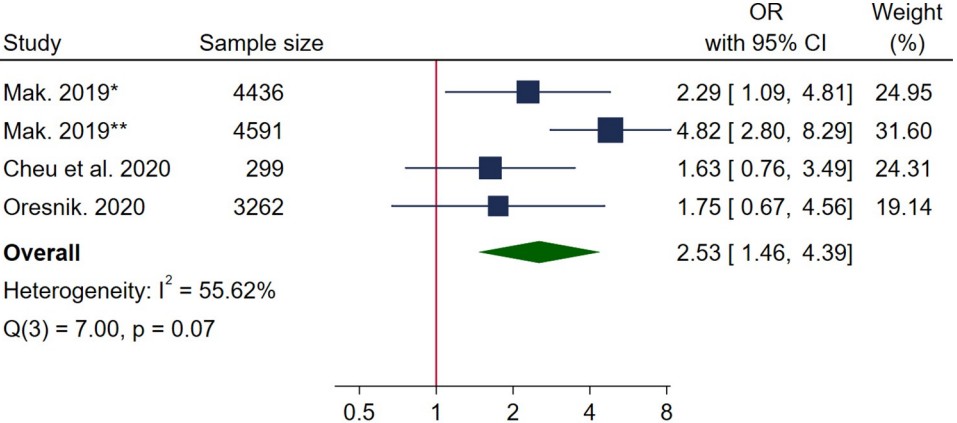

**Fig 5. Forest plot for meta-analysis of the association between mood disorder and food insecurity.** Measurement of FI: Cheu and colleagues, 2020 used USDA HFSSM 10 items [50]; Mak, 2019 used USDA HFSSM 10 items [68]: Oresnik, 2020 used USDA HFSSM 18 items and 1 single item adapted from USDA HFSSM [69].

CI [1.31, 4.04]) [54], whereas one reported no association between marginal food security and CES-D depression scores of >21 or >24 (AOR 2.97, 95% CI [0.85, 10.35, and 3.64], 95% CI [0.77, 17.31], respectively) [63]. Four studies reported a consistent positive association between depressive symptoms and varying degrees of food insecurity (ARRs ranging from 1.72, 95% CI [1.40, 2.11 to 3.26], 95% CI [2.46, 4.32]) [58]. Higher mean depressive symptom scores were reported by 2 studies for women with acute or chronic food insecurity compared with food secure (mean 11.10 (SD 7.80), 13.60 (SD 8.80), and 9.30 (SD 7.40), respectively, $p < 0.001$) [56], and for food insecure women compared with food secure women (mean 0.81 (SD 0.46), mean 0.49 (SD 0.32), respectively, $p < 0.001$) [53]. One study reported positive associations between food insecurity and feeling more depressed than usual due to the pandemic (APR 2.32, 95% CI [2.13, 2.53]) [60].

Six studies reported data on anxiety-related outcomes [49,53,56,59,68,69], all showing significant positive associations for women experiencing food insecurity. One study reported higher generalised anxiety in both acute and chronic food insecure groups compared with food secure (mean 6.00 (SD 5.50), 6.60 (SD 4.50), and 4.60 (SD 4.50), respectively, $p < 0.001$) [56]. Two studies found higher odds of diagnosed anxiety disorder among women experiencing food insecurity compared to food secure (AOR 2.49, 95% CI [1.09, 5.67]) [69] and among women experiencing moderate-severe food insecurity (AOR 3.23, 95% CI [1.92, 5.43]) but not for marginal food insecurity (AOR 1.82, 95% CI [0.94, 3.54]) [68]. Two studies reported a higher anxiety score for women experiencing food insecurity compared to food secure (MD 1.55, 95% CI [1.04, 2.05] [49] and mean 2.04 (SD 0.51), mean 1.71 (SD 0.32), respectively, $p < 0.001$) [54]. One study reported a positive association between food insecurity and feeling more anxious than usual due to the pandemic (APR 1.79, 95% CI [1.71, 1.88]) [59].

Other outcomes reported relating to food insecurity and mental health included increased common mental disorders (IRR 1.90, 95% CI [1.30, 2.80]) [61], and mental health symptoms (β 0.54, $p < 0.001$ reported in a path analysis) [53]. One study reported that women experiencing moderate-severe food insecurity had increased odds of poor/fair perceived mental health than food secure (AOR 3.79, 95% CI [1.52, 9.48]) and no association for marginal food insecurity (AOR 2.23, 95% CI [0.64, 7.78]) [68]. This study also reported no association between marginal and moderate-severe food insecurity and poor/fair

perceived health generally (AOR 1.60, 95% CI [0.60, 4.26, and 1.52], 95% CI [0.82, 2.80]), or perceiving their health to be worse than the previous year (AOR 0.96, 95% CI [0.38, 2.39, and 0.91], 95% CI [0.54, 1.53]) [68]. There were poorer levels of resilience among food insecure women reported, including feeling no love, dissatisfied with life, despair and loss of control (ORs ranging from 2.24, 95% CI [1.38 to 3.65 to 9.33] 95% CI [3.39, 25.69]) [29], but no association between marginal and moderate-severe food insecurity and having a weak sense of community (AOR 1.08, 95% CI [0.72, 1.62, and 1.33], 95% CI [0.92, 1.92]), [68] or community status (β −0.02, 95% CI [−0.10, 0.02]) [54].

## Food insecurity and infant health outcomes

Thirteen studies reported associations between food insecurity and infant health outcomes, including birth weight (*n* = 4) [50,53,55,56], low birth weight or SGA (*n* = 5) [29,30,48,50,67], high birth weight or LGA (*n* = 5) [30,48,50,52,67], preterm delivery (*n* = 7) [29–31,48,50,57,64], admission to neonatal intensive care unit (NICU) (*n* = 3) [30,50,51], and other neonatal morbidity outcomes (*n* = 4) [30,48,50,57] (see S9 Table). Meta-analysis was possible for birth weight, SGA, LGA, preterm delivery, and NICU admission.

Three studies reported data for birth weight; 2 were high quality [53,55] and 1 medium quality [56] with a pooled sample size of 594 women. Four studies reported data for SGA and LGA, 3 high quality [30,50,67] and 1 medium quality [48] with a pooled sample size of 12,739 women. Pooled results showed no association between food insecurity and mean birth weight (MD −58.26 g, 95% CI [−128.02, 11.50], $I^2$ 38.41%) (Fig 6), SGA (OR 1.20, 95% CI [0.88, 1.63]) $I^2$ 44.66%) (Fig 7), or LGA (OR 0.88, 95% CI [0.70, 1.12] $I^2$ 11.93%) (Fig 8). Additional data reported that could not be pooled in the meta-analysis (see S9 Table) showed no association with food insecurity and birth weight for gestational age z-score [53] or low birth weight (RR 1.21, *p* = 0.51 and OR 1.39, 95% CI [0.45, 4.29]) [29,50]. However, one study reported a positive association between food insecurity and high birth weight in 2 different adjusted models: AOR 2.07 (95% CI [1.07, 3.98]) when adjusting for child sex, race, ethnicity, parent age, education, income, and study site; and AOR 1.96 (95% CI [1.01, 3.82]) when adding poor neighbourhood food environment to the model (S9 Table) [52].

Seven studies reported data for preterm delivery [29–31,48,50,57,64]. Five studies could be pooled in a meta-analysis, 3 high quality [30,50,64] and 2 medium quality [31,48] with a pooled sample size of 18,880. Pooled analysis showed no association with food insecurity (OR

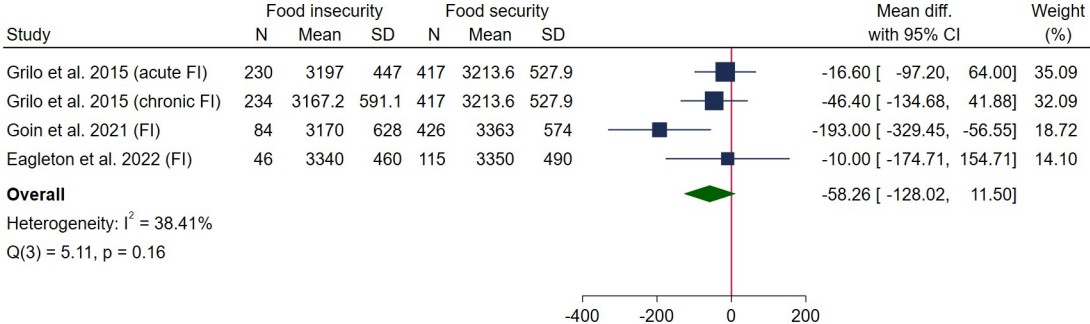

**Fig 6. Forest plot for meta-analysis of the association between birth weight and maternal food insecurity (mean difference).** Legend: Categories of food insecurity and food security: food secure vs. food insecure Goin and colleagues, 2021 [55], Eagleton and colleagues, 2022 [53]; acute and chronic food insecurity Grilo and colleagues, 2015 [56]. Measurement of FI: Grilo and colleagues, 2015 [56] used 1 screening question; Goin and colleagues, 2021 [55] and Eagleton and colleagues, 2022 [53] used USDA HFSSM.

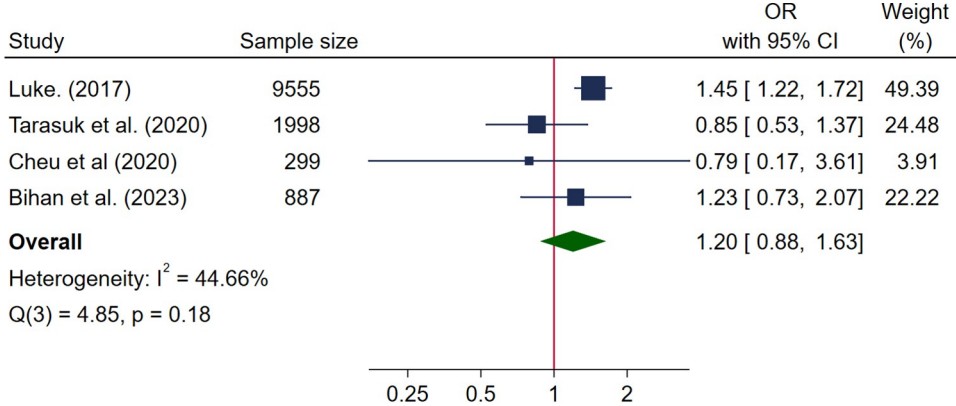

**Fig 7. Forest plot for meta-analysis of the association between small for gestational age and food insecurity.**
Legend: Categories of food insecurity and food security: food secure vs. food insecure, Luke, 2017 [67]; food secure vs. marginal food insecurity/moderate or severe food insecurity, Tarasuk and colleagues, 2020 [30]; adequate food security vs. inadequate food security, Cheu and colleagues, 2020 [50]. Measurement of FI: Luke, 2017 [67] used 1 question rapid assessment; Cheu and colleagues, 2020 [50] used USDA HFSSM 10 items; Tarasuk and colleagues, 2020 [30] used USDA HFSSM 18 items; Bihan and colleagues, 2023 [48] used 1 single item adapted from USDA HFSSM.

1.18, 95% CI [0.98, 1.42], $I^2$ 0.00%) (Fig 9). The 2 studies that could not be pooled in meta-analysis also showed no association between food insecurity and preterm delivery (RR 1.27, $p$ = 0.45 and ARR 1.04 95% CI [0.72, 1.51]) [29,57]. One study additionally reported that preterm delivery differed by parity, where parous women experiencing food insecurity had significantly increased odds of preterm delivery (OR 1.41, 95% CI [1.04, 1.91]) compared to nulliparous women [31].

Three high-quality studies reported data for NICU admission and could be pooled in the meta-analysis [30,50,51] with a pooled sample size of 2,367 women. Pooled analysis showed no association with food insecurity (OR 2.01, 95% CI [0.85, 4.78], $I^2$ 70.48%) (Fig 10). Four studies

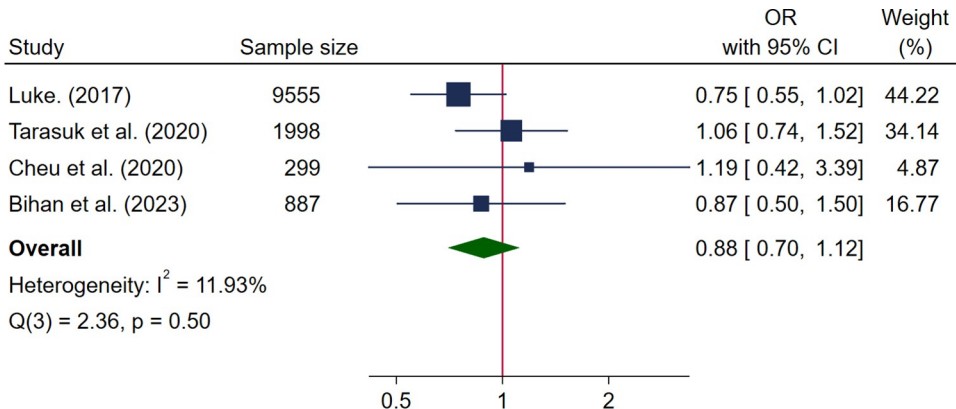

**Fig 8. Forest plot for meta-analysis of the association between large for gestational age and food insecurity.**
Legend: Categories of food insecurity and food security: food secure vs. food insecure, Luke, 2017 [67]; food secure vs. marginal food insecurity/moderate or severe food insecurity, Tarasuk and colleagues, 2020 [30]; adequate food security vs. inadequate food security, Cheu and colleagues, 2020 [50]. Measurement of FI: Luke, 2017 [67] used 1 question rapid assessment; Cheu and colleagues, 2020 [50] used USDA HFSSM 10 items; Tarasuk and colleagues, 2020 [30] used USDA HFSSM; Bihan and colleagues, 2023 [48] used 1 single item adapted from USDA HFSSM.

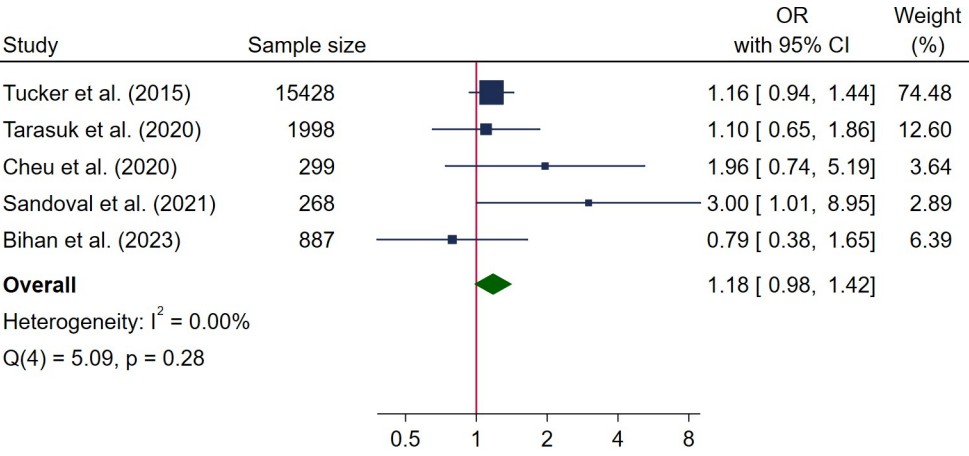

**Fig 9. Forest plot for meta-analysis of the association between pre-term delivery and food insecurity.** Legend: Categories of food insecurity and food security: food secure vs. food insecure, Tucker and colleagues, 2015 [31]; food secure vs. marginal food insecurity/moderate or severe food insecurity, Tarasuk and colleagues, 2020 [30]; adequate food security vs. inadequate food security, Cheu and colleagues, 2020 [50]; food security vs. prenatal household food insecurity, Sandoval and colleagues, 2021 [64]; not food insecure vs. food insecure, Bihan and colleagues, 2023 [48]. Measurement of FI: Cheu and colleagues, 2020 [50] used USDA HFSSM 10 items; Tarasuk and colleagues, 2020 [30] used USDA HFSSM; Bihan and colleagues, 2023 [48] used 1 single item adapted from USDA HFSSM; Tucker and colleagues, 2015 [31] used 1 screening question; Sandoval and colleagues, 2021 [64] used USDA HFSSM 6 items.

reported additional neonatal morbidity outcomes [30,48,50,57]. Studies reported no association between congenital anomalies and marginal (ARR 1.96, 95% CI [0.99, 3.86]) or moderate-severe food insecurity (ARR 1.13, 95% CI [0.52, 2.45]) [30]. Additionally, no association was reported between food insecurity and shoulder dystocia (OR 1.52, 95% CI [0.06, 37.49]) [48], respiratory distress (OR 1.36, 95% CI [0.29, 6.38]) [50], 5-min Apgar score <7 (OR 0.80, 95% CI [0.09, 6.45]) [50], glucose <40 mg/dl in first 24 h (OR 0.88, 95% CI [0.19, 4.01]) [50], or neonatal hypoglycaemia (OR 1.49, 95% CI [0.31, 7.06] [50] and OR 0.85, 95% CI [0.24, 2.97] [48]). Inconsistent patterns were reported for food insecurity and mortality: one study

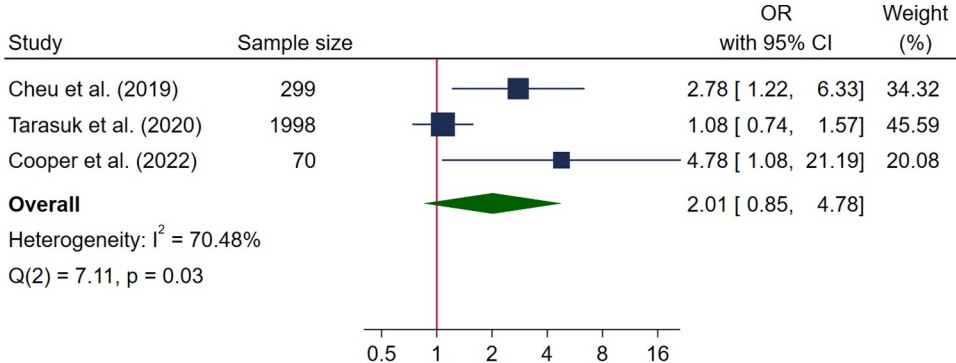

**Fig 10. Forest plot for meta-analysis of the association between admission to NICU and food insecurity.** Legend: Categories of food insecurity and food security: adequate food security vs. inadequate food security, Cheu and colleagues, 2020 [50]; food secure vs. food insecure, Cooper and colleagues, 2022 [51]; food secure vs. marginal food insecurity/moderate or severe food insecurity, Tarasuk and colleagues, 2020 [30]. Measurement of FI: Cheu and colleagues, [50] 2020 used USDA HFSSM 10 items; Tarauk and colleagues, 2020 [30] used USDA HFSSM; Cooper and colleagues, 2022 [51] used USDA HFSSM.

reported a positive association between food insecurity and stillbirth (ARR 2.71, 95%CI [1.13, 6.45]) [57] and one reported no association between food insecurity and neonatal death/stillbirth (OR 0.91, 95% CI [0.04, 19.06]) [48].

## Discussion

This systematic review has extensively explored associations between women experiencing food insecurity in HICs post the 2008 global financial crisis, and pregnancy health outcomes for women and their infants. The meta-analysis identified positive associations between food insecurity and GDM, high stress levels, and mood disorder, but no association with birth weight, SGA, LGA, preterm delivery, NICU admission, or cesarean delivery. Narrative synthesis also identified patterns for associations between food insecurity and some maternal physical health outcomes including dental health, and serum PFOS, but evidence was limited for infant health outcomes. Further, there were highly consistent data for associations between food insecurity and a range of maternal mental health outcomes that could not be included in meta-analysis due to a limited number of studies reporting similar enough data for pooling. These included increased risks for depression, anxiety, and mental health disorders, and women experiencing food insecurity being more likely to also experience stress events, depressive symptoms, poor resilience, and lower perceived mental health.

The associations between food insecurity and maternal mental health outcomes could be a consequence of the compounded effect from intersecting and interconnected inequalities such as gender and class. In previous studies, women have described that due to a lack of income they are unable to participate in normal consumer routes for purchasing food, or participate socially, and describe feelings of tension between their nutritional desires and their inability to meet them in addition to ongoing uncertainty around access to food [23]. Pregnancy, birth, and early parenthood represent life-changing moments. Moments of change can instigate a period of poorer mental health or worsen preexisting mental health conditions [71]. Therefore, a concern is that women with food insecurity experience more severe mental health challenges (double burden on mental health) as they experience the mental health costs associated with food insecurity in addition to the potential mental health costs from pregnancy. Maternal mental health must be acknowledged, not only to improve maternal health and well-being but also because children exposed to maternal mental ill health are at increased risk of experiencing psychological and developmental disturbances [72,73]. The findings from this review make it evident that screening for food insecurity during pregnancy and mental health conditions in HICs is necessary to ensure care for mother and baby. The World Health Organisation (WHO) recognises the importance of screening, diagnosing, and managing perinatal mental health conditions, publishing new guidelines for integrating perinatal mental health into maternal and child health services in late 2022 [71]. Specific to poverty, WHO recommend health services monitor infants for malnutrition and forge links with community providers that provide services addressing the wider social determinants of health, such as housing associations. Further, they encourage women to form strong social networks that provide support during financially difficult times, a sentiment that will be difficult to achieve given the challenges of living with poverty, mental health, and pregnancy.

The finding that pregnant women experiencing food insecurity have significantly increased odds of GDM is concerning as developing GDM can result in poorer health outcomes for both mother and baby. These include LGA infants, complications at birth, and increased life course risk of developing T2DM for women and their children [74]. Children born to women with GDM could be at risk of metabolic disease later in life [75], thus food insecurity during pregnancy can have important potential life course implications for the child. Risk factors for

GDM include maternal overweight and obesity. Women experiencing food insecurity during pregnancy are significantly more likely to be living with obesity [76] which in itself increases the risk of obesity development in the child [77]. Therefore, food insecurity compounds the potential increased risks for the child relating to both maternal obesity and GDM. Potential mechanisms underpinning the association between food insecurity and GDM include the wider determinants of maternal obesity such as housing conditions and food environment. Accounts of food insecurity report poor kitchen facilities and utensils to cook and store healthy meals [23]. Further, food insecurity is associated with energy-dense diets and living in obesogenic environments, thus increasing the risk of developing obesity and GDM [78,79]. Another potential mechanism explaining the association between food insecurity and GDM could be the dysregulated eating patterns characteristic of food insecurity which could alter metabolic regulation contributing to the development of GDM [80]. Eating patterns for those experiencing food insecurity are typified by food being consumed when food is available, and restricted when food shortages exist [23]. Cycles of food abundance at the beginning, then food shortages at the end of the month have been reported [81]. Thus, charitable food aid, social support, or income distributed weekly rather than monthly could help mitigate these eating patterns. A study has also shown food insecurity is indirectly linked with higher body mass index via maladaptive coping mechanisms such as emotional eating [82], which this review has identified could be triggered by significant high levels of stress women experience. Finally, among women, food insecurity is associated with nutritionally poor diets, lacking fruit and vegetables which could deprive food insecure women of the protective nutrients required to prevent disease such as GDM [83–85].

This review identified limited data for infant health outcomes, and data that were included showed a lack of association between food insecurity and SGA, LGA, preterm delivery, and NICU admission. Some of these data are in conflict with evidence from LMICs showing that low birth weight was higher in food insecure groups compared to food secure groups [86–88], potentially due to differences in context between LMICs and HICs relating to food insecurity and availability of types of foods. However, our meta-analysis was limited by the limited number of studies reporting infant outcomes and the context being in North America, which may have impacted on the lack of significance in pooled results despite an overall pattern towards positive directions of effect. The USA context may bias the results due to a potential intervention effect from the well-established Special Supplemental Nutrition Program (SNAP) for Women, Infants and Children (WIC) in the USA [89]. Evidence shows that WIC reduces food insecurity and can improve pregnancy outcomes such as gestational hypertension [90], preterm delivery [91], SGA and NICU admission [92,93]. However, other HICs do not have embedded support in place like WIC, thus further research in other contexts is needed to fully understand the impact of food insecurity on pregnancy outcomes.

## Strengths and limitations

There are several strengths to this systematic review including following gold standard methods such as registering the study protocol in advance of conducting the review on PROSPERO, using a rigorous multistranded search strategy to supplement our database searches with forwards and backwards citation chaining, independent duplicate screening, data extraction and quality assessments, and contacting study authors to overcome potential publication bias. As our review was holistic in terms of pregnancy outcomes, we did not include any specific outcome terms in our search strategy as this may have omitted studies reporting relevant maternal and infant pregnancy outcome data that were not in a predefined list of outcome search terms. Further, the use of the PRISMA and MOOSE guidelines helps to improve the transparency

and scientific merit when reporting systematic reviews and meta-analysis [32,94,95]. While we were not able to conduct statistical analysis for publication bias as we had originally planned due to insufficient studies pooled in each meta-analysis, there were multiple included studies reporting nonsignificant results. This suggests that the potential for publication bias relating to nonsignificant results was minimal.

There are also several limitations to the review. We identified a small number of studies reporting consistent outcomes or reporting data in a way that could be pooled, thus limiting our ability to conduct meta-analysis. We made an a priori decision to conduct meta-analysis if there were 3 or more studies reporting similar enough exposures of food insecurity and pregnancy outcomes, resulting in 9 meta-analyses reported in this review. However, there were wide variations in the sample sizes of included studies overall and in the pooled sample sizes for the meta-analysis. These ranged from 594 women for birth weight meta-analysis to 19,440 women for high stress meta-analysis; this potentially contributed to some of the heterogeneity observed in analysis. Additionally, as sample size influences the weighting of any individual study, the inclusion of one large study with multiple smaller studies will impact on the weighting towards a single study. Understanding the sources of heterogeneity was particularly relevant for maternal high stress. Although we had planned to explore sources of heterogeneity using meta-regression methods, this was not possible as there were fewer than 10 studies included. While the overall effect size of this meta-analysis should be interpreted with caution due to the potential influence of one study and wide confidence intervals, there was a consistent positive association between food insecurity and stress in all studies reporting these outcomes. The narrative synthesis also identified maternal mental health to be consistently significantly associated outcome with food insecurity across all papers reporting these related outcomes. The ability to conduct more robust meta-analysis for maternal mental health would enable us to draw more robust and generalisable conclusions about the extent that food insecurity is associated with maternal mental health and explore contributing factors and sources of heterogeneity between studies. Another source of heterogeneity limiting this review was the use of different food insecurity measures between studies. This meant we were unable to compare the different food insecurity levels between studies, thus, impacting the interpretation of the association of food insecurity on different health outcomes across included studies. Further, meta-regression was not possible; we were therefore unable to explore the influence of studies adjusting for confounding variables.

Of the included studies none were classified as low quality, and the majority of studies contributing data to meta-analysis were high quality. All studies included in meta-analysis for stress, NICU, mood disorders were high quality, with 1 or 2 medium quality studies contributing to GDM, LGA, SGA, birth weight, cesarean, and preterm delivery meta-analysis. A low response from study authors upon requesting additional information, such as frequency data to enable calculation of ORs and 95% CIs, impacted the meta-analysis as some studies were not able to be included. Further, although there were some studies that reported adjusted models for associations between food insecurity and pregnancy outcomes, there was inconsistency in the factors included in the adjusted models. Many studies did not include adjustments at all, and there may be issues with residual confounding for unmeasured, or poorly measured variables. While it would be unethical to control for this in studies of food insecurity, for example, by using randomisation methods, thorough consideration must be given to residual confounding given the intersecting nature of inequalities and the likelihood that food insecurity does not exist in isolation of other factors that may also be influencing the pregnancy outcomes.

The use of different tools to assess food insecurity and the use of different food insecurity categories, most notably in classification of persons between the marginal or moderate versus low or very low level of food security presents a limitation to this review as it introduces study

heterogeneity. The majority of included studies conducted in North America used the validated USDA HSFFM or its short/modified version. The USDA HFSSM was developed in the USA and has been used to monitor prevalence of food insecurity annually since 1995 [96]. The use of its different forms (short forms or single item) may introduce classification inconsistencies as the use of a short form, or a single item, is not as rigorous a measurement of the severity of household food insecurity. For example, in Australia, a single-item measure of food insecurity which was incorporated in their National Health Survey only tends to capture marginal food insecurity [97]. In addition, the prevalence of food insecurity is commonly estimated on the basis of the number of "food insecure" conditions reported (in the form of affirmative responses) on a food insecurity scale [98]. While the USA and Canada share the same food insecurity measurement, the HFSSM, they use different classification system for different level of food insecurity severity [99]. The USA requires 3 affirmative responses before classifying a household as food insecure, "food secure" is classified if there is no affirmative response or up to 2 affirmative responses to questions on the HFSSM, "low food security" if there are 3 to 7 and "very low food security" if there are 8 or more [100]. Whereas, in Canada, one affirmative response on either the adult or the child scale is taken as indicating food insecurity (marginal food security level) [100–102]. Thus, when applied to the same data, the USA's classification system would yield a lower figure for food insecurity and may have less sensitivity to detect vulnerability to negative health consequences associated with food insecurity than the Canadian system [101,103]. Therefore, there is a need for more consistent measures to facilitate between study comparisons. In addition, given the impact of food insecurity on maternal physical and mental health outcomes, it is important to use the full-scale measure of food insecurity and consider the marginal level of food insecurity to better understand the relationship between food insecurity and health outcomes.

Finally, almost all the studies included in this review were conducted in North America, primarily the USA ($n$ = 20), with 3 from Canada and 2 from Europe (1 France, 1 United Kingdom). This highlights the gap in the literature investigating the impact of food insecurity in pregnancy across HICs. Unlike in North America where the food aid system is institutionalised, and research has been conducted for many years, in European HICs it was not until after the global financial crisis of 2008 and the parallel rise of charitable food aid that food insecurity has become an urgent issue, with increasing prevalence widely impacting on health status and systems such as maternity. This gap also limits the generalisability of this review's results to other HIC contexts. Importantly, there may be differences in the impact of food insecurity in pregnancy in other HIC contexts where there is a high prevalence of food insecurity and different, or absent, support mechanisms embedded within public health and maternity services.

## Conclusions

This review has identified that food insecurity in pregnancy is significantly associated with some poorer maternal health, particularly relating to GDM and mental health. However, little data were available outside of the North America context and thus a key recommendation from this review is for further research in wider HIC contexts to fully explore potential associations, especially in countries with high prevalence of food insecurity. Findings of this review are important to inform the improvements in maternity services to better be able to support women experiencing food insecurity, particularly relating to a need for screening for food insecurity in conjunction with GDM and mental health. Importantly, these results identify potential life course implications of experiencing food insecurity in pregnancy for both women's health and their children's health, alongside increasing prevalence of food insecurity in many HICs. This requires urgent public health interventions to tackle poverty and evaluation

should factor in the potential benefits of alleviating food insecurity in pregnancy on the short- and long-term health and well-being of women and their children.

## Supporting information

**S1 File. 1a PRISMA and 1b MOOSE checklists.**
(DOCX)

**S1 Table. Translation of search terms across databases.**
(DOCX)

**S2 Table. Grey literature website searches.**
(DOCX)

**S3 Table. Details of contact with study authors.**
(DOCX)

**S4 Table. Adapted Newcastle–Ottawa scale for quality assessment for cohort study.**
(DOCX)

**S5 Table. Table of included studies additional characteristics.**
(DOCX)

**S6 Table. Quality assessment for included studies.**
(DOCX)

**S7 Table. Association between food insecurity and maternal physical health in pregnancy outcomes.**
(DOCX)

**S8 Table. Association between food insecurity and maternal mental health outcomes.**
(DOCX)

**S9 Table. Association between food insecurity and infant health outcomes.**
(DOCX)

## Acknowledgments

We would like to thank Ms. Hannah Mehmood for being part of the team carrying out title and abstract screening for this review. We would also like to thank authors Professor Barbara Laraia and Dr. Maddy Power who responded to our request for additional information.

## Author Contributions

**Conceptualization:** Nicola Heslehurst.

**Data curation:** Zoë Bell, Giang Nguyen, Gemma Andreae, Stephanie Scott, Letitia Sermin-Reed, Amelia A. Lake, Nicola Heslehurst.

**Formal analysis:** Zoë Bell, Giang Nguyen, Gemma Andreae, Amelia A. Lake, Nicola Heslehurst.

**Investigation:** Zoë Bell, Giang Nguyen, Gemma Andreae, Stephanie Scott, Letitia Sermin-Reed.

**Methodology:** Nicola Heslehurst.

**Project administration:** Nicola Heslehurst.

**Validation:** Zoë Bell, Giang Nguyen, Amelia A. Lake, Nicola Heslehurst.

**Visualization:** Zoë Bell, Giang Nguyen.

**Writing – original draft:** Zoë Bell, Giang Nguyen.

**Writing – review & editing:** Zoë Bell, Giang Nguyen, Gemma Andreae, Stephanie Scott, Letitia Sermin-Reed, Amelia A. Lake, Nicola Heslehurst.

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
