## [Editor Report · Decision Letter 0]

11 Apr 2024

Dear Dr Heslehurst, 

Thank you for submitting your manuscript entitled "Associations between food insecurity in high-income countries and pregnancy outcomes: a systematic review and meta-analysis" for consideration by PLOS Medicine.

Your manuscript has now been evaluated by the PLOS Medicine editorial staff as well as by an academic editor with relevant expertise and I am writing to let you know that we would like to send your submission out for external peer review.

Please re-submit your manuscript within two working days, i.e. by Apr 15 2024 11:59PM.

Kind regards,

Katrien G. Janin, PhD

Senior Editor

PLOS Medicine

---

## [Decision Letter · Decision Letter 1]

21 May 2024

Dear Dr. Bell,

Thank you very much for submitting your manuscript "Associations between food insecurity in high-income countries and pregnancy outcomes: a systematic review and meta-analysis" (PMEDICINE-D-24-01101R1) for consideration at PLOS Medicine. 

As you will see, the reviewers were positive about the paper but, they raised a number of questions about specific study details and the methodological approach. After discussing the paper with the editorial team and an academic editor with relevant expertise, I’m pleased to invite you to revise the paper in response to the reviewers’ comments. We plan to send the revised paper to some of all of the original reviewers*, and of course we cannot provide any guarantees at this stage regarding publication. 

When you upload your revision, please include a point-by-point response that addresses all of the reviewer and editorial points, indicating the changes made in the manuscript and either an excerpt of the revised text or the location (eg: page and line number) where each change can be found. Please submit a clean version of the paper as the main article file and a version with changes marked should as a marked-up manuscript. Please also check the guidelines for revised papers at http://journals.plos.org/plosmedicine/s/revising-your-manuscript for any that apply to your paper. 

We ask that you submit your revision by the 11th of June. However, if this deadline is not feasible, please contact me by email, and we can discuss a suitable alternative. 

Please don’t hesitate to contact me directly with any questions (kjanin@plos.org). If you reply directly to this message, please be sure to ‘Reply All’ so your message comes directly to my inbox. 

We look forward to receiving your revised manuscript. 

Sincerely,

Katrien Janin, PhD

PLOS Medicine

plosmedicine.org

*Please note: If your article is accepted, you may have the opportunity to make the peer review history publicly available. The record will include editor decision letters (with reviews) and your responses to reviewer comments. If eligible, we will contact you to opt in or out. 

Comments from the Academic Editor:

1. The authors use fixed effects models when the I2 indicates high heterogeneity. The high level of heterogeneity in the meta-analytical summary statistics can be dealt with using random effects (please also see ther eveiwer comment on this)

2. Food insecurity will - obviously - not exist in isolation. There will be myriad associated factors. The language of causality needs to be removed given the potential for some of the associations to be explained by other factors associated with food insecurity but not fully corrected for in the analyses (i.e. residual confounding).

3. Focus on the results where the associations are clearly positive and the 95% CI exclude unity. It is ok to mention trends but it should be accompanied by discussion of the 95% CI and the acknowledgement that the data are also consistent with no effect.

1) Inline with the Academic Editor and the statistical reviewer (see below), we also think the use of fixed effects models in the face of high heterogeneity is questionable. The high level of heterogeneity in the meta-analytical summary statistics can be dealt with using random effects (it could also be good to see both the FE and RE estimates).

2) Minor request: We wonder if perhaps it is best to more explicit about the fact that most of the studies are not just from North America, but from the US (also to be added in the Abstract as a limitation maybe?)

General editorial request:

1) Abstract layout:

Please structure your abstract using the PLOS Medicine headings (Background, Methods and Findings, Conclusions). 

2) Author summary: 

At this stage, we ask that you include a short, non-technical Author Summary of your research to make findings accessible to a wide audience that includes both scientists and non-scientists. The authors summary should consist of 2-3 succinct bullet points under each of the following headings: 

• Why Was This Study Done? Authors should reflect on what was known about the topic before the research was published and why the research was needed. 

• What Did the Researchers Do and Find? Authors should briefly describe the study design that was used and the study’s major findings. Do include the headline numbers from the study, such as the sample size and key findings. 

• What Do These Findings Mean? Authors should reflect on the new knowledge generated by the research and the implications for practice, research, policy, or public health. Authors should also consider how the interpretation of the study’s findings may be affected by the study limitations. In the final bullet point of ‘What Do These Findings Mean?’, please describe the main limitations of the study in non-technical language. 

The Author Summary should immediately follow the Abstract in your revised manuscript. This text is subject to editorial change and should be distinct from the scientific abstract. Please see our author guidelines for more information: https://journals.plos.org/plosmedicine/s/revising-your-manuscript#loc-author-summary

3) Discussion layout:

Please present and organize the Discussion as follows: a short, clear summary of the article's findings; what the study adds to existing research and where and why the results may differ from previous research; strengths and limitations of the study; implications and next steps for research, clinical practice, and/or public policy; one-paragraph conclusion.

4) Statistical reporting 

Wherever applicable, please quantify the main results with 95% CIs and p values. When reporting p values please report as <0.001 and where higher as p=0.002, for example. When reporting 95% CIs please separate upper and lower bounds with commas instead of hyphens as the latter can be confused with reporting of negative values. 

5) Supplementary materials: 

Please note that supplementary materials are not checked and will be posted as supplied by the authors. Therefore, please double check. Please cite your Supporting Information as outlined here: https://journals.plos.org/plosmedicine/s/supporting-information - Please note you may use almost any description as the item name of your supporting information as long as it contains an "S" and number. For example, “S1 Appendix” and “S2 Appendix,” “S1 Table” and “S2 Table. Please ensure each supplementary material has a call out (link) from your main manuscript. 

Comments from the reviewers:

Reviewer #1: Paper is much improved. 

Reviewer #2: Alex McConnachie, Statistical Review

The paper by Bell et al presents a systematic review and meta-analysis of observational studies looking at the association between food insecurity during pregnancy and maternal and birth outcomes. This review looks at the use of statistics in the paper.

These are generally fine. Basic meta-analysis methods are appropriately applied. Other planned analyses, using meta-regression, sensitivity analyses, and looking at publication bias, were not done, due to the relatively small number of studies involved in each meta-analysis; this is in line with general recommendations, so is acceptable. I have a few comments, though these are fairly minor.

At various places in the paper, the authors lapse into language that implies a causal link between food insecurity and outcomes. For example, in the abstract, lines 36-37, "Meta-analysis showed that food insecurity significantly increased high stress level", or in the first sentence of the conclusions, "This review has identified that food insecurity has substantial impacts on maternal health and wellbeing". As with any observational research, however plausible the causal links may be, this sort of language should generally be avoided. I can tell that the authors are aware of this, from the discussion of limitations, and I know how hard it can be to get the wording right at times, but I suggest a little more editing is required.

Similarly, I know how hard it can be to avoid the overuse of "significant" or "significantly" when reporting results. In this paper, however, the authors have a tendency to use phrases such as "non-significant positive association", which is a slight problem. If an association is not (statistically) "significant", then there is no evidence of an association. Is it then appropriate to highlight whether each non-association is positive or negative? I think I understand what the authors are trying to do; even though many associations are not statistically significant, there is a general pattern among the point estimates, with a tendency for poorer outcomes amongst those with food insecurity. I think this is ok to highlight, as part of the discussion, but flagging every estimate in the text of results section may be going a little too far.

Only fixed effects analyses are presented. Given the high levels of heterogeneity, and for some analyses, the large weight applied to a single study, some might argue that random effects analyses would be preferred. Perhaps both results could be presented? Otherwise, I think that more justification for the choice of only using fixed effects methods is required.

In the methods section, there is a reference to R Shiny being used for forwards and backwards citation chaining. I thought R Shiny was a package for building web applications. Would it be more accurate to say that citation chaining was done using an app developed in R Shiny? Or have I misunderstood?

Very minor points

Line 372-373: why no confidence interval?

Line 384-385: the CI looks very asymmetrical, when I would expect it to be symmetrical for a beta coefficient.

Reviewer #3: This is a well-conducted and well-written systematic review and meta-analysis on an important and highly relevant topic (the effect of food insecurity on maternal and infant outcomes), given the cost-of-living crisis and financial pressures faced by high-income countries. It is a re-submission of a paper that I have reviewed previously, and the authors have addressed my concerns. They do a particularly good job of discussing the complexities of analysing these associations using observational data, whilst also emphasising the strength of the evidence for poor maternal and infant outcomes in affected populations. I also note that their searches have been updated to include studies up to November 2023. I recommend that the paper is suitable for publication. 

Minor comments:

In the PROSPERO record, it states that random effects models will be used for the meta-analysis if the I² is more than 75%. However, fixed-effects models are used throughout (even with an I² of 94.6% for high stress levels). In the version of the paper that I reviewed previously, random effects models were used. The conclusions do not change between the versions (unless new studies have been added), but it would be good to add a sentence to the manuscript justifying the use of fixed-effects models and any sensitivity analyses conducted (to reassure readers that these do not change the conclusions).

In the reporting of the results, "significant" or "non-significant" are used throughout. I do not know the journal's policy on this (but know that other journals discourage the use of these terms). 

I would recommend that the phrasing is changed where the confidence intervals for the pooled ratio estimates cross 1, from "non-significant positive association" (for example, in lines 279-281, "non-significant positive association between food insecurity and caesarean section (OR 1.18 95% CI 0.85, 1.64") to "no association" (or the journal's preferred phrasing).

Lisa Hurt

Cardiff University

[LINK]

---

## [Decision Letter · Decision Letter 2]

16 Jul 2024

Dear Dr. Bell,

Thank you very much for re-submitting your manuscript "Associations between food insecurity in high-income countries and pregnancy outcomes: a systematic review and meta-analysis" (PMEDICINE-D-24-01101R2) for review by PLOS Medicine.

I have discussed the paper with my colleagues and the academic editor and it was also seen again by the reviewers. I am pleased to say that provided the remaining editorial and production issues are dealt with we are planning to accept the paper for publication in the journal.

[LINK]

If you have any questions in the meantime, please contact me (kjanin@plos.org) or the journal staff on plosmedicine@plos.org.  

We look forward to receiving the revised manuscript by Jul 23 2024 11:59PM.   

Sincerely,

Katrien Janin, PhD

Senior Editor 

PLOS Medicine

plosmedicine.org

Requests from Editors:

Thank you for your detailed response to the editors' and reviewers' comments. I have discussed the paper with my colleagues and the academic editor, and it has also been seen again by the original reviewers. The changes made to the paper were satisfactory to the reviewers and the academic editor. 

I have just a few minor editorial comments and reporting style alteration request for you before I can move to issue an editorial acceptance of your manuscript.

1. Thank you for providing your PRISMA checklist. When completing the checklist, please use section and paragraph numbers, rather than page numbers. (For example: Methods, paragraph 2). Please also add the following statement, or similar, to the Methods: "This study is reported as per the Preferred Reporting Items for Systematic Reviews and Meta-Analyses (PRISMA) guideline (S1 Checklist)." Feel free to add here aswell that you are also have provided a MOOSE (Meta-analyses Of Observational Studies in Epidemiology) Checklist (and add the appropriate Sx link to your manuscript).

2. Statistical reporting : As per standard PLOS Medical requirements, please provide 95% CIs and p values for all results where appropriate (including the abstract). We suggest reporting statistical information in the following format: ‘x’; (95% CI [‘y’,’ z’] p value) and use commas as opposed to hyphens (as these can be confused with negative values) to separate upper and lower bounds. For p values, please report as p<0.001 and where higher as 'p=0.002'. Please add the statistical method used to your method section. We also invite you to report p values to consistently to the third decimal digit - thousandths.

3. ACKNOWLEDGMENTS/ DECLARATIONS

Please remove all statements apart from acknowledgements, author contributions and abbreviations from the end of the main manuscript and include these only in the relevant parts of the manuscript submission form. Funding, competing interest, and data availability will be compiled as metadata.

4. Please cite the reference numbers in square brackets. Citations should precede punctuation.

5. References: Where website addresses are cited, please specify the date of access (e.g. [accessed: 12/06/2023]). - E.g see reference 70. Please check and amend thorough. 

See https://journals.plos.org/plosmedicine/s/submission-guidelines#loc-references for further details on reference formatting. 

GENERAL COMMENTS:

6. Supplementary materials: Please note that supplementary materials are not checked and will be posted as supplied by the authors. Therefore, please double check. Please cite your Supporting Information as outlined here: https://journals.plos.org/plosmedicine/s/supporting-information - Please note you may use almost any description as the item name of your supporting information as long as it contains an "S" and number. For example, “S1 Table” and “S2 Table". Please ensure each supplementary material has a call out (link) from your main manuscript. 

7. Social media: To help us extend the reach of your research, if you not have already done so please provide any X (formerly known as Twitter) handle(s) that would be appropriate to tag, including your own, your co-authors’, your institution, funder, or lab. Please enter in the submission form any handles you wish to be included when we post about this paper.

Thank you very much and I look forward to receiving your updated manuscript!

Katrien

Comments from Reviewers:

Reviewer #2: Alex McConnachie, Statistical Review

I thanks the authors for their consideration of my original points, and I have no further comments.

One point of clarification: the missing confidence interval is now on line 399 of the revised manuscript: "mental health symptoms (β 0.54, p<0.001)". Confidence intervals are given for other beta coefficients, so I assume this is just an oversight.

Reviewer #3: This is the second revision of this paper that I have reviewed. It is an important and timely systematic review. The authors have addressed the comments of all reviewers thoroughly. I recommend that the paper is published.

Lisa Hurt, Cardiff University

[LINK]

---

## [Editor Report · Decision Letter 3]

24 Jul 2024

Dear Dr Bell, 

On behalf of my colleagues and the Academic Editor, I am pleased to inform you that we have agreed to publish your manuscript "Associations between food insecurity in high-income countries and pregnancy outcomes: a systematic review and meta-analysis" (PMEDICINE-D-24-01101R3) in PLOS Medicine.

PRESS

Sincerely, 

Katrien G. Janin, PhD 

Senior Editor 

PLOS Medicine